

# Geomagnetic field declination: from decadal to centennial scales

Venera Dobrica[1], Crisan Demetrescu[1], Mioara Mandea[2]

[1]Institute of Geodynamics, Romanian Academy, Bucharest, Romania
[2]Centre National d'Etudes Spatiales, Paris, France

*Correspondence to*: Venera Dobrica (venera@geodin.ro)

**Abstract.** Declination annual means time-series longer than a century provided by 24 geomagnetic observatories world-wide, together with 5 Western European reconstructed declination series over the last four centuries have been analyzed in terms of frequency constituents of the secular variation at inter-decadal and sub-centennial time-scales of 20-35 and, respectively, 70-90 years. Observatory and reconstructed time-series have been processed by several types of filtering,

namely Hodrick-Prescott, running averages, and Butterworth. The Hodrick-Prescott filtering allows to separate a quasi-oscillation at decadal time scale, supposed to be related to external variations and called '11-year constituent', from a long-term trend. The latter has been decomposed in two other oscillations, called 'inter-decadal' and 'sub-centennial' constituents by applying a Butterworth filtering with cutoffs at 30 and 73 years, respectively. The analysis shows that the generally accepted geomagnetic jerks occur around extrema in the time derivative of the trend and coincide with extrema in the time

derivative of the 11-year constituent. The sub-centennial constituent is traced back to 1600, in the five 400-year long time-series, and shows to be a major constituent of the secular variation, geomagnetic jerks included.

## 1 Introduction

The temporal variation of the geomagnetic field has been monitored for decades mainly by continuous recordings in geomagnetic observatories. In spite of the growing number, their geographical coverage is highly uneven. Since the longest

recorded series of observations at geomagnetic observatories do not exceed some 150 years, research has seen increasing interest in historical spot measurements to construct as long as possible time-series of geomagnetic elements (declination and inclination) going back over some centuries (Malin and Bullard, 1981 for London area; Cafarella et al., 1992 a, b for Rome; Barraclough, 1995 for Edinburgh; Alexandrescu et al., 1996, 1997 and Mandea and Le Mouël, 2016 for Paris area; Korte et al., 2009 for the Munich area). Korte et al. (2009) also included archeomagnetic data, in order to infer information going

back to 1400 AD. Such an interest has also been present in Eastern Europe (Bucha, 1959 for the Czech and Slovak territories; Valach et al., 2004 for Slovakia; Atanasiu, 1968; Constantinescu, 1979; Soare et al., 1998 for Romania), reconstructions go back to 1850 and included intensity elements of the geomagnetic field. Let us also note the collection of declination and inclination data from sea voyages over the 15-18th centuries by Jonkers et al. (2003), and the spherical harmonics (SH) model by Jackson et al. (2000). This model includes these data and describes the geomagnetic field



evolution since 1590, for all geomagnetic elements (the intensity values are based on an assumed uniform dipole decay rate before 1850).

The last four decades – since the finding by Courtillot et al. (1978) of 'geomagnetic jerks' – have seen a focus of research on the features of the time evolution of the geomagnetic field originating in the Earth's core (main field).

Geomagnetic jerks are viewed phenomenologically as sharp changes, within one year or so, in the temporal variation of the main field (secular variation), expressed as the first time derivative of the geomagnetic field time-series, or steps in the secular acceleration expressed as the second time derivative. The definition of geomagnetic jerks is usually illustrated by declination (D) or by the eastward horizontal component of the field (Y) thought to be the least influenced by external variations. Alexandrescu et al. (1997) and Korte et al. (2009) explored the possibility that jerks also occurred before the

modern era for which they were evidenced. A review on the subject has been published by Mandea et al. (2010). The geomagnetic jerks have been treated as chaotic fluctuations of the field by Qamili et al. (2013). Brown et al. (2013) have been interested in the sharp time changes in the secular variation, using an improved method to account for the external variations in data (Wardinski and Holme, 2011) and for a jerk identification (Pinheiro et al., 2011); they find by far more events than had previously been determined, with a general trend of increased number of identified jerks towards the end of

the 20th century and beginning of the 21st century. The new satellite era offers high resolution magnetic data and the possibility to deeply investigate these events. Indeed, Finlay et al. (2015) and Chulliat et al. (2015) indicate the occurrence, in 2006, 2009 and 2012, of geomagnetic secular acceleration pulses at the core surface. We can note as dates of these new events 2003 (Olsen and Mandea, 2007), 2007 (Olsen et al., 2009; Chulliat et al., 2010), 2011 (Chulliat and Maus, 2014), 2014 (Torta et al., 2015; Kotze, 2017).

Periodicities have been found in the evolution of the geomagnetic field. Fourier spectral analysis (e.g. Currie, 1973; Alldredge, 1977; Langel et al., 1986), but also the Maximum Entropy Method (Jin and Thomas, 1977), the Empirical Mode Decomposition (Roberts et al., 2007; Jackson and Mound, 2010), as well as calculations of torsional waves in the Earth's core (Zatman and Bloxham, 1997; Dickey and de Viron, 2009; Buffett et al., 2009), pointed to periodicities at several time-scales, such as ~11, 13-30, 50-60, and 60-90 years. Buffett (2014) explores the MAC (Magnetic, Archimedes, Coriolis

forces) waves and their role in generating long-period variations (60 years) of the field. Periods shorter than the solar cycle time-scale have also been detected, both in some geomagnetic field models (Silva et al., 2012) and in observatory data (e.g. Ou et al., 2017). These studies revealed periodicities in the time domain of about 4-7 years and, respectively 2-3 years. The latter was considered of external origin, while the former has been used as an argument in favour of a six-year internal variation. We note, however, that both periodicities are very close to the harmonics of the solar cycle, inducing the idea of

their external origin.

Demetrescu and Dobrica (2014) demonstrate the presence, in 24 observatory time-series (annual means of the D, H, Z components over some 100-150 years) of constituents of secular variation at time-scales of 22- and ~80-year, superimposed on a so-called steady variation. A slightly different time scale, of 70-75 years, not of ~80 years as seen in H and Z data, seemed to be characteristic in case of declination. They also show that these constituents are seen in the first time



derivative of the field, too. The episodes of increasing and decreasing secular variation that result from the superposition of the two constituents are separated by a smooth transition that lasts several years; the sharpness of a geomagnetic jerk is decided by the external effects still existing in data. Indeed, the external effects, mainly controlled by the 11-year solar cycle, are still present when averaging available data to obtain the annual means. These remaining external contributions are

present mostly in the intensity elements of the recorded field (H, Z), and affect less the declination (e. g., Olsen and Mandea, 2007). The effects of external contributions in studying the secular variation have been emphasized and quantitatively shown for the European observatories by Verbanac et al. (2007), in terms of correcting annual means using information on external sources, by Wardinski and Holme (2011), in terms of a stochastic (covariant) modeling method, and by Dobrica et al. (2013), in terms of secular variation maps. Recent studies by Gillet et al. (2010, 2015) and Holme and de Viron (2013) point to a

possible ~6 year variation originating in the core. A direct modeling of torsional waves (Cox et al., 2016) shows, however, that a 6-year wave in the core cannot give the estimated effects at Earth's surface, placing the problem of internal high-frequency signals under debate. Demetrescu and Dobrica (2014) tentatively show that the ~80-year variation can be traced back to the 15$^{th}$ century, using three long time-series of declination, of London (Malin and Bullard, 1981), Rome (Cafarella et. al., 1992 a, b), and Munich (Korte et al., 2009).

In the present paper we focus on declination data and revisit 24 time-series of observatory data, updating the available measurements with additional ones since 2007, the last year included in a previous analysis (Demetrescu and Dobrica, 2014). Additional data allow us to better constrain the 1999 (Mandea et al., 2000) and 2007 (Chulliat et al., 2010) events and to infer a possible external contribution to the 2003 geomagnetic jerk (Olsen and Mandea, 2007). Here, new methods are applied in filtering the time-series and novel approaches regarding the quasi-periodicities of the constituents at

longer time-scales are considered (Hodrick and Prescott (1997)). Additionally, a special attention is given to the 11-year solar-cycle-related constituent, present in the declination annual means. Finally, we elaborate on our previous analysis of three very long declination time-series, by *i)* including in the analysis two more: Paris (Alexandrescu et al., 1996, 1997; Mandea and Le Mouël, 2016) and Edinburgh (Barraclough, 1995), *ii)* discussing the first time derivative of the five time-series, *iii)* comparing in detail our analysis on jerk occurrence to the Alexandrescu et al. (1997) and Korte et al. (2009) ones,

and *iv)* comparing with time-series provided by the *gufm1* main field model by Jackson et al. (2000).

## 2 Data

### 2.1 Observatory data

Annual means of declination as given by http://www.geomag.bgs.ac.uk/data_service/data/annual_means.shtml have been used. The locations of the 24 observatories with 100-150 years long time-series, labelled with their IAGA codes are shown

in Fig. S1 (Supplementary material), superimposed on the WMM2010 declination map at the geomagnetic epoch 2010.0 (http://www.ngdc.noaa.gov/geomag/WMM/data/WMM2010/WMM2010_D_MERC.pdf). In Fig. 1 we show, as an example, time-series of declination and of its first time derivative at Niemegk (Germany) observatory (IAGA code NGK).



## 2.2 Historical data

Five long time-series referenced above, for Edinburgh, London, Paris, Munich, and Rome have been considered in the present study. Data used, in the order of decreasing latitude of the location, are as follows:

- Edinburgh: raw data published by Barraclough (1995), adjusted to Eskdalemuir observatory (ESK);
- London: raw data published by Malin and Bullard (1981), adjusted to Hartland observatory (HAD);
- Paris: raw data published by Alexandrescu et al. (1996, 1997) and reviewed recently by Mandea and Le Mouël (2016), adjusted to Chambon la Foret observatory (CLF);
- Munich: 11-year filtered smoothing spline fitted to raw data, as published by Korte et al. (2009), adjusted to Furstenfeldbruck observatory (FUR);
- Rome: assembled time-series using data published by Cafarella et al. (1992 b) for Rome area and for three successively operating Italian observatories (Pola, 1881-1922; Castellaccio, 1933-1962; L'Aquila, 1960-2011), adjusted to L'Aquila observatory (AQU).

These time-series are used in Section 4.

## 3 Method

A Fourier spectral analysis (FFT) on the 24 declination time-series (Fig. 2a) and of their time derivative (Fig. 2b) shows a broad spectral peak at around 73 years (60-100) that dominates by far other (broad) peaks at ~30, ~22, and ~17 years. At this stage we also remark differences between observatories regarding frequencies corresponding to these lines, which are commented upon below. Some of these differences could arise from the different lengths of the time-series, as some tests (not shown here) of repeating the FFT for the same time-series truncated to different lengths seems to indicate. Demetrescu 20 and Dobrica (2014) notice that dominating powerful signals at larger time-scales in data tend to contaminate the filtered time-series meant to show quasi-periodic variations at smaller time-scales. That is why in the present paper we apply a Hodrick and Prescott (1997) (HP) type analysis, which is able to separate oscillatory features at smaller (e. g. decadal) time-scales from trends representing variations at larger (e. g. centennial) time-scales.

The HP filter separates a time-series $y_t$ into a trend component $T_t$ and a cyclical component $C_t$ such that $y_t = T_t + C_t$.
The function for the filter has the form

$$\sum_{t=1}^{m} C_t^2 + \lambda \sum_{t=1}^{m} \left[ (T_t - T_{t-1}) - (T_{t-1} - T_{t-2)} \right]^2$$

where $m$ is the number of samples and $\lambda$ is the smoothing parameter. The first sum minimizes the difference between the time-series and its trend component (which is its cyclical component). The second sum minimizes the second-order difference of the trend component (which is analogous to minimization of the second derivative of the trend component). If 30 the smoothing parameter is 0, no smoothing takes place. As the smoothing parameter increases in value, the smoothed series



becomes more linear. Appropriate values of the smoothing parameter depend upon the data sampling. In our case data being yearly sampled we apply a smoothing parameter of 100, recommended by Hodrick and Prescott (1997) and checked by us after a few tests with λ varying between 10 and 1600 with a step of 50 (not shown here). The HP filter is equivalent to a cubic spline smoother, with the smoothed portion in $T_t$.

Variations at larger time-scales seen in the trend given by HP filtering have been further decomposed in two other oscillations, by applying a Butterworth (1930) filtering with certain cutoffs corresponding to (quasi)periods of ~73 and 30 years, as indicated by two broad peaks in the superimposed FFT spectra of the trend for all 24 observatories shown in Fig. S2 (Supplementary material).

### 3.1 Observatory data

We have applied the described methods on observatory and historical data. With respect of observatory data, firstly we show in Fig. 3, as an example of data processing, results for Niemegk. The first time derivative of declination (the first differences of annual means) is shown in the upper plot, together with the trend given by the HP filter. The cyclic component is also plotted (middle). No difference exists when the latter is compared to the superimposed time-series obtained as residuals of filtering the original time-series with an 11-year running average window or with a high-pass 11-year cutoff Butterworth

(1930) filter. Verbanac et al. (2007) show that the residual signal after removing CM4 core field model from the annual averages of European observatories has a clear solar-cycle signature and can be modeled down to ± 2 nT, using magnetospheric ring current data (Dst) and an ionospheric field proxy (Ap). Wardinski and Holme (2011) characterize the external effects stochastically, analyzing the correlated so-called 'noise' in the time-series of X, Y, and Z components at world-wide distributed observatories and the Dst index in a first step and residuals at Niemegk instead of Ap, in a second

step. We note that core sources could also contribute to the high-frequency 11-year signal and its constituents, for instance by the 6-year signal discussed by Gillet et al. (2010, 2015) and Holme and de Viron (2013). We tend to attribute the 11-year signal to external sources, based on a long list of papers referenced by Demetrescu and Dobrica (2014).

The lower plot of Fig. 3 shows a comparison between the cyclic component and a two-waves sinusoidal fit for two time intervals, namely 1890-1960 and 1960-2011, based on different simulations. It can be observed the strong presence of

harmonics of an 11-year cycle, superimposed on an ~11-year oscillation in the first part of the time-series and on a significant ~22-year oscillation over the last ~40 years. This behaviour is quite different from that of the recorded geomagnetic field intensity, in which the external effect in annual means appears as a clear 11-year variation along the entire time-series (not shown, see Demetrescu and Dobrica (2014)).

Since the HP filtering applied to the trend is not able to further separate it in constituents, we appeal (a) to running

averages (Demetrescu and Dobrica, 2014) and (b) Butterworth (1930) filtering to get the time-series corresponding to the ~73 and at 25-35 years time-scales. In Fig. 4 we show, again as an example, the results for NGK. In the first case, the constituents of the trend are obtained by successively smoothing the trend time-series with 30- and 73-year running average and subtracting them from the trend time-series and, respectively, from the 30-year average time-series. A similar result is



obtained in the second case, of using 30- and, respectively, 73-year cutoff Butterworth filtering. Both methods have advantages and disadvantages. As one can notice in Fig. 4, the running average filtering produces time-series that include the full information from the unfiltered data on a certain central portion, but no information for both ends of time-series (the 30- and the 73-year smoothed time-series are shorter by 15 and 36 years respectively, at each end), while the Butterworth filter

produces time-series of the same length as unfiltered data, but with distorted amplitudes at ends. The dates of maxima and minima in the filtered time-series are, however, correctly retrieved, allowing conclusions on (quasi)periodicities in data to be drawn for the entire time interval covered by data. Let us note that if we have used instead of 30- and 73-year filtering on trend values the 22- and 80-year filtering by Demetrescu and Dobrica (2014), we would have obtained similar results (extrema at the same moments, but slightly different amplitudes), as one can see in Fig. 4. Of course, any pair of constituents

one chooses, 30- and 73- or 22- and 80-year ones, the sum of the two constituents is the same, namely the trend plotted in the top panel of Fig. 3.

       Let us discuss the differences between observatories, seen in frequencies corresponding to the broad spectral lines singled out above (Fig. 2 and Fig. S2 (Supplementary material)). The information regarding the actual periodicities at various observatories would not be lost when adopting a certain average value (e. g. 73 or 80 years) in data processing.

Indeed, unless the window in the running average or the cutoff value in the Butterworth filtering is a multiple of the hidden period in data, the filtered cyclical component is itself a cyclical component of the same period as the original component (Appendix, Demetrescu and Dobrica, 2014). This can be seen in Fig. 4, where time-series obtained using values of 22 and 30 years or, respectively, 80 and 73 years for filtering data are compared. In case of H and Z (not shown) the filtered ~80-year constituent shows an oscillation of that period (actually a mean period of 78 years), but in case of D a slightly different time-

scale, of 70-75 years seems to be characteristic. Since D is a non-linear function of the field vector, one should not expect that field oscillations induced by the core sources be identical in H and D (Roberts et al., 2007), so a slight difference might occur. Also, according to the same authors, one should not expect exactly the same response at all observatories to variations in the core sources changes. Consequently, in the remaining of this paper we use the terms "inter-decadal" and "sub-centennial" constituents for the ones at the 20-35 and, respectively, 70-90-year time-scales.

25       We remark here that, at odds with the internal inter-decadal and sub-centennial constituents, the 11-year constituent is very noisy, on one hand because errors in the annual means (measurement noise, baseline definition, changes in pillars etc.) are retained almost entirely in this time-series, and, on the other hand, as a result of the time derivative operator that enhances noise and brings forward harmonics of the 11-year constituent that are not significant in data (compare also Figs. 2a and 2b). Besides, the solar cycle length variability (between 8 and 14 years over the past 10 cycles) also contributes to the

noise in the high-passed 11-year time-series. Superimposing spectra of the cyclic component for the 24 declination time-series (Fig. S3 (Supplementary material)), the noisiness is evident. However, in spite of that, some specific lines can be distinguished:

       - lines in the 15-25 year interval, corresponding to the ~22-year constituent, detected in the last ~40 years of the cyclic component time-series of Fig. 3;



- lines in the 8-14 year domain, corresponding to the 11-year constituent;

- lines in the 4-7 and 2-3 year domains, corresponding to the first two harmonics of the 11-year constituent. We note that the 4-7 year signal covers the 6-year signal detected in variations of length-of-day (Holme and de Viron, 2013) and in wave processes discussed by Gillet et al. (2010, 2015),  pointing to a possible core contribution to the observed variation. We also note the presence of stronger peaks in the 2-3 years period domain than those in the 4-7 years one. This observation is in line with the study by Ou et al. (2017).

We are aware of the fact that, due to noise, the separation in frequency domains is not an ideal one. For instance, there are large peaks occurring between frequencies corresponding to periods between 7 and 9 year; they characterize spectra for Hartland and, with smaller amplitude, Canberra observatories. We consider this possibly linked to the less precise values at the beginning of the time-series, noise that is retained in the 11-year signal and can significantly alter the FFT.

### 3.2 Historical data

The historical data have been processed as it has been done for observatory time-series. Since data are sparser and sparser before ~1800, for London, Paris, Rome, and Edinburgh a cubic B-spline interpolation of early data is used to obtain a plot with continuous annual values, after removing the evident outliers. The latter can be a result of less precise measurements and/or less precise reduction to the location of the present-day observatory, contributing with data over the most recent periods. Due to the temporal distance of several years between historical data at the beginning of the reconstructed time-series, the spline line might show artefact wiggles. For that part of time-series a linear or quadratic interpolation could be more appropriate. The Munich series has been already filtered by Korte et al. (2009).

Fig. 5 shows the time derivative of the Paris time-series together with the superimposed HP trend; the inter-decadal and sub-centennial constituents of the trend are shown too. The noise problem becomes stronger when the time derivative of the declination series is considered. The time derivative enhances, as expected, short time variations presented at decadal and shorter time-scales by less accurate historical data before ~1850, that results in higher amplitudes than for the observatory era of the two constituents of the HP trend (compare 0.12 to 0.025 and 0.05 to 0.01 in case of the inter-decadal, and respectively, sub-centennial constituents).

### 4 Results and discussion

### 4.1 Observatory data

The two constituents of the secular variation for the 24 observatories considered in this study, as obtained by a HP filtering, namely the trend and the decadal cyclic variations, are shown in Fig. 6. The trends are referred to the average value for the time interval in which they are defined. The two constituents of the trend, the inter-decadal and the sub-centennial variations, as obtained by a Butterworth filtering, are also plotted. We superimpose the time-series from the 24 observatories,



corresponding to each of the time-scales, in order to emphasize common features and differences. We also indicate in Fig. 6 the accepted occurrence time of geomagnetic jerks (e.g. Mandea et al., 2010; Brown et al., 2013).

The cyclic constituent obtained from a HP filtering (second top panel in Fig. 6), supposed to show the effects of external sources in data, is very noisy and prevents in this form any interpretation regarding this constituent of the recorded secular variation. However, plotting only data from the considered European observatories (Fig. 7), emphasizes the strong presence of harmonics of the 11-year cycle, superimposed on the 11-year oscillations in the first part of the time-series and on the significant 22-year oscillation in the last ~40 years (see also Fig. 3, NGK). We have chosen geographically close European observatories in order to enhance the visual effect, as they are affected by the same 22- and ~80-year variations (see Demetrescu and Dobrica, 2014) and show similar time evolutions. The similarity of the 11-year signal in European observatories is also noticed in a previous paper (Dobrica et al., 2013).

The 2003 jerk shown in Fig. 6 has been evidenced for limited areas only ($\Delta dY^2/dt^2$ slightly negative over Central and Eastern Europe and positive along the 90-100°E meridian, noted by Olsen and Mandea (2007)). From Fig. 7 it seems that the external effects might play a role in characterizing the 2003 jerk, as a 5-year running averages smoothing, meant to get rid of variations related to the first harmonic of the 11-year variation, attenuates the sharp variation of the declination time derivative seen in the raw data (upper plot). We note that the smoothing would attenuate also the 6-year component discussed by Gillet et al. (2010, 2015) and Holme and de Viron (2013), if present in data. According to the same figure, other recent European geomagnetic jerk occurrence dates should be slightly shifted by one year, from 1999 and 2007, to 1998 and 2006 respectively, when extrema actually occur in the 11-year variation. However, due to the filtering procedure and the annual means sampling of data, the events by the end of series might not be so well described. It is worth to note that in terms of analysis shown in Figs. 6 and 7, the very recent geomagnetic jerks occurred in 2011 and 2014, evidenced for limited areas by Chulliat and Maus (2014) and Torta et al. (2015) in Atlantic sector and Atlantic and European sectors, respectively, reveal a strong influence of the decadal constituent at those dates. A more detailed analysis of the 11-year constituent of the secular variation is beyond the scope of this paper.

In Figs. 6 and 7 the vertical lines mark epochs of accepted geomagnetic jerks; they occurred around extrema in the time derivative of the trend variation, produced by a combination of the two constituents, at ~30 and ~73-year timescales, and coincide with extrema in the time derivative of the external variation (with the above observation, regarding the 1999 and 2007 events). Phase differences between individual time-series explain differences in the occurrence time and geographical distribution of geomagnetic jerks. Once the external contributions to the first differences of the observatory annual means, of comparable amplitude with the trend variations are minimized, the core contribution to the observed secular variation no longer exhibits the very sharp appearance of geomagnetic jerks. According to our quantitative analysis of recorded data presented in Figs. 6 and 7 (see also Demetrescu and Dobrica (2014)), the geomagnetic jerk might be seen as a result of a more general phenomenon, namely the evolution of the secular variation as a result of a superposition of two (or several) waves describing effects of processes in the Earth's core at two (or several) time-scales. This is in line with Bloxham et al. (2002) and Alldredge (1984; 1985) who have advanced this possibility, based on core flow modeling




arguments and, respectively, on geometric arguments. Finlay and Jackson (2003) and Jackson and Finlay (2007) have identified core surface equatorial westward moving magnetic flux patches that can be either a result of core flow vortices entrained by a larger scale westward flow, or Alfvén waves excited in the core. That the flow in the core is turbulent became common consideration with many studies on geodynamo and core flow modelling from secular variation data (see the

review by Holme (2015) on the latter). The turbulent flow, as inferred by De Santis et al. (2003) by looking at the power spectra of H, Z, D annual means time-series, does not exclude but, on the contrary implies the existence of vortices with various time and space scales. The "waves" we speak of above could be in fact surface manifestations of core surface vortices that move around and survive for a given timespan, as shown by Demetrescu and Dobrica (2014). The latter discussed the map appearance of their steady, 22-year and ~80-year variations, pointing to the different space scales of the

three ingredients that manifest themselves at the three timescales. As noted by Holme (2015), the core flow models existing to date, including those of higher resolution based on satellite models of the field and secular variation, are not able to predict small scale features, as field models cannot resolve details in the core field smaller than the spherical harmonic degree 13.

## 4.2 Historical data

Demetrescu and Dobrica (2014) have previously analyzed three of the long time-series of historical magnetic declination data (London, Munich, Rome) and showed that the sub-centennial variation is present back in time to the 15[th] century. Here, we define and characterize the sub-centennial variation in case of the secular variation of declination for five available time-series.

The time-series showing the declination at the five locations are plotted in Fig. 8. We also superimpose time-series

obtained from *gufm1* (Jackson et al., 2000) for the corresponding location. The spot measurements before observatory era are affected by much larger errors than observatory measurements, due, on one hand, to the equipment accuracy, and on the other to the non-corrected external variations and/or reduction to the location of the present-day observatory. As mentioned in Section 2, for London, Paris, Rome, and Edinburgh a cubic B-spline interpolation of early data is used to obtain a plot with continuous annual values. The spline curve shows artifact wiggles before ~1700. For that part of time-series a linear or

quadratic interpolation could be more appropriate to describe the long-term evolution. The Munich time-series, published by Korte et al. (2009), had been smoothed with an 11-year filter and a 2.5-year knot space spline. Fig. 9 shows, for the five locations, only the first time derivative of the HP trend, together with the corresponding HP trend time derivative in *gufm1* values.

In the following, only the sub-centennial constituent, less affected by noise than the decadal and inter-decadal

constituents, is discussed. The five curves in Fig. 10, showing the sub-centennial constituent of the trend plotted in Fig. 9, demonstrate that the latter is not restricted to the last 150 years. So does also the sub-centennial constituent derived from *gufm1*. The same conclusion, based on Empirical Mode Decomposition applied to Munich data, has been invoked by





Jackson and Mound (2010). The latter also found a longer period, of 160 years. Referring to the appearance of the curves in Figs. 8 and 9, a variation at a much larger timescale, of 400 years or longer, could, however, be present in data.

Several maxima and minima are evident in the sub-centennial constituent time derivative plots (Fig. 10) before 1900, besides the known ones over the 20[th] century. Comparing the five time-series of the sub-centennial constituent time

derivative with each other and with the corresponding *gufm1* model, a few interesting observations could be emphasized:

- Firstly, all curves as derived from data and/or model show the same maxima and minima of the sub-centennial constituent after 1850, namely maxima at 1850-1880, 1920-1930, and minima at around 1900 and 1960;

- Secondly, the amplitude of the sub-centennial constituent before and after 1900 seems to be comparable, in spite of the lower quality of data for the first 300 years of the time-series. Larger amplitude variations at the start of the time-series

stem probably from the poorly constrained data in that time interval;

- Finally, before 1850 the noise in data is more evident in case of London time-series. However, a synchronous maximum around 1800 and a minimum around 1820 can be seen in London, Paris, and Rome curves, but not in the Munich and in the *gufm1* ones, that show only an inflexion at 1790-1800 and at ~1820 (change to a lower secular acceleration – the second time derivative of the field – and then change again to a higher secular acceleration). Another maximum, well

developed around 1750 in Paris, Munich, and Rome plots, as well as in all *gufm1* ones, is not seen in the London curve. Back toward 1650, another maximum seems to be present in London and Paris curves, but not in the Munich one, while the *gufm1* plots indicate its presence. We might account this behavior on the temporally less dense data before 1780 and on the spline smoothing.

In terms of geomagnetic jerks, Alexandrescu et al. (1997), based on a synthetic declination curve for Paris inferred

from Paris and London series, recognized one event around 1870 in annual and monthly means time-series, also detected in Helsinki, Furstenfeldbruck and Oslo data. Other possible events are noted around 1700, 1730, 1750, 1770 and 1785. These dates correspond, within a few years, with maxima and minima of the sub-centennial variation time derivative plotted in Fig. 10 (1690, 1740, 1780, 1800, 1850-1860, 1900). Data prior to 1870 were considered by Alexandrescu et al. (1997) too noisy or unreliable to clearly reveal geomagnetic jerks, at least of comparable amplitude with the 1870, 1901, 1925, 1969, and

1978 jerks. However, as mentioned above, the amplitude of the sub-centennial variation before and after 1900 is comparable. Korte et al. (2009) compared the Munich smoothed secular variation time-series with a time-series of the same length (1400-2000) for Paris, containing archeomagnetic and other measured data prior to 1700, adjusted to CLF, spline-smoothed in the same way as the Munich time-series. A good agreement characterizes the time interval 1770-2000 and, as the authors stated, surprisingly consistent secular variation and acceleration between the smoothed curves from the two locations is found for

the time span 1400-1580, in spite of the rather low quality of data over this time-span. Significant differences between the two locations exist however between 1580 and 1770. Korte et al. (2009) also note that in both curves the time interval 1765-1865 seems to be devoid of strong rapid secular changes.

When comparing the possible geomagnetic jerks (called "events" by Korte et al. (2009)) in the Munich curve, with the maxima and minima of the first time derivative of the sub-centennial constituent for the same location, plotted in Fig. 10,





we find that most of them (1448, 1508, 1558, 1693, 1741, 1861, 1889, 1932) coincide within 0-3 years with maxima and minima of our sub-centennial constituent (1460, 1510, 1560, 1690, 1740, 1780, 1800, 1850-1860, 1900). In the time interval 1765-1865, considered to be devoid of strong rapid secular changes by Korte et al. (2009), our analysis detects, as mentioned above, inflections at 1790-1800 and at ~1820, which are close to the 1790-1810 maximum and, respectively, to the 1818-1828 minimum seen in the London, Paris, and Rome curves.

Considering these results we can suggest that geomagnetic jerks are only a part of a variation at a much longer timescale, the sub-centennial constituent. A certain contribution, most visible over the last 40 years, comes also from the 20-30-year inter-decadal constituent. The larger noise in data before 1900 prevents a possible identification of the latter at earlier times, the only variation that could be observed being the sub-centennial one.

# 5 Conclusions

Our results underline the importance of the time perspective one has on geomagnetic data: besides the contribution of the sub-centennial constituent in defining geomagnetic jerks, what we called 'steady variation', based on 150 years of observatory data (Demetrescu and Dobrica, 2014), proves to be only a part of a larger timescale variation, when 400 years of data are available.

Declination annual means time-series longer than a century provided by 24 geomagnetic observatories world-wide, together with 5 reconstructed declination series over the last four centuries in the Western Europe have been analyzed in terms of frequency constituents of the secular variation at inter-decadal and sub-centennial time-scales of 20-35 and, respectively, 70-80 years. Observatory time-series until 2015 have been processed by several types of filtering, namely Hodrick-Prescott, running averages and Butterworth. Average windows of, and respectively cutoffs at 11, 30, and 73 years have been used to account for broad lines in the FFT spectra corresponding to (a) the external solar-cycle-related contamination in the annual averages, the so-called 11-year or *decadal* constituent, to (b) a 20-35-year constituent, named *inter-decadal*, and, respectively, to (c) a broad intense spectral line (60-100 years) present in data, the so-called *sub-centennial* constituent, singled out by the HP filtering and FFT analysis of the constituents. The average values used in filtering to obtain the variations of the three constituents of the observed time derivative of declination have no consequences on the evolution and dominant period of the retrieved constituents at individual observatories. This is expected, as any filtered cyclical component is itself a cyclical component of the same period as the original one. Also, a slight difference between average sub-centennial timescales in declination and in the vector components of the field could be noticed (73 years compared to 78 years). These results confirm the conclusion by Demetrescu and Dobrica (2014), based on shorter sub-centennial and inter-decadal time-series.

The accepted geomagnetic jerks occur around more pronounced extrema in the time derivative of inter-decadal constituent and coincide with extrema in the time derivative of the 11-year constituent (except the '1999' and the '2007' events). Around 1925, 1969, and in 2006 the extrema in the sub-centennial constituent coincide in time or is close to the





extrema in the inter-decadal constituent, leading to more pronounced geomagnetic jerks. Phase differences between individual time-series explain differences in the occurrence time, geographical distribution and magnitude of geomagnetic jerks. Once the external contributions to the first differences of the observatory annual means – of comparable amplitude with the inter-decadal and sub-centennial constituents – are minimized, the observed secular variation no longer exhibits a

clear V-shape at time of geomagnetic jerks.

The detected extrema in the historical data have been compared with events interpreted in terms of geomagnetic jerk occurrence dates proposed by other authors (Alexandrescu et al., 1997; Korte et al., 2009). It appears that possible "events" in jerk terms, at 1700, 1730, 1750, 1770, 1785, considered with a question mark by Alexandrescu et al. (1997) because of too noisy or unreliable data, are occurring close to maxima and minima of the sub-centennial constituent. The sub-centennial

constituent has comparable amplitudes before and after 1900, in spite of lower quality of data in the first 300 years of the analyzed time-series, making it a reliable tracer of geomagnetic jerks in the past. Unfortunately, because of noise in the reconstructed time-series, the inter-decadal variation, a constituent of the secular variation, could not be recovered and complete information on these phenomena occurrence is limited.

According to our results, epochs of geomagnetic jerks may vary as much as a couple of years from one series to

another. Over the investigated period, some very clear long periods exist between two successive and well-defined jerks. Over this long-term tendency, less well-defined events can be observed, as the many noted since magnetic satellite data are available (e.g. Torta et al., 2015). We suggest that the geomagnetic jerk concept should be considered as a more general notion, namely the evolution of the secular variation as a result of superposition of two (or more) constituents describing effects of processes in the Earth's core at two (or more) time-scales. Revealing the causes of these variations from the point

of view of mechanisms in the core is beyond the scope of this work.

**Acknowledgements**

The data used in this paper are freely available at http://www.geomag.bgs.ac.uk/ data_service/data/annual_means.shtml. The historical geomagnetic declination has been compiled from papers by S. R. C. Malin and E. C. Bullard, L. Cafarella, A. De Santis, and L. Meloni, D. R. Barraclough, M. Alexandrescu, V. Courtillot and J.-L. Le Mouël, and M. Korte, M. Mandea and

J. Matzka (op. cit). The study has been done in the frame of the project IDEI-UEFISCDI 93/2011. Partial results were presented at MagNetE5 (Rome, 2011) and IUGG (Melbourne, 2011) meetings. We are indebted to Lili Cafarella (INGV-Rome) who kindly provided historical geomagnetic data for Italy, and not in the least, to anonymous geomagnetic observatory staff and to the World Data Centers on Geomagnetism for obtaining and, respectively, keeping data used in this study.



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

30





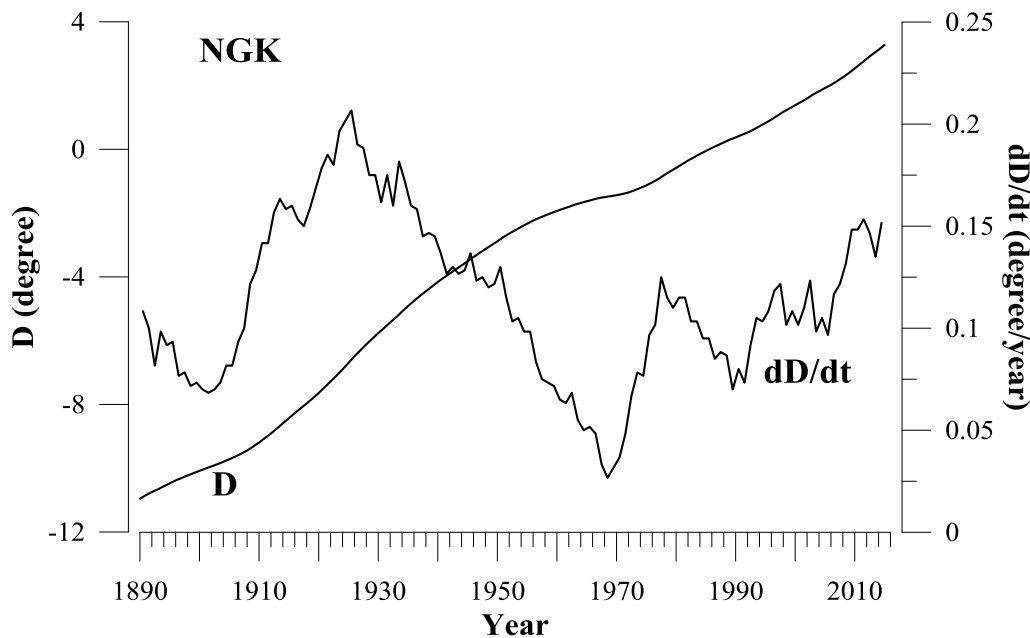

**Figure 1: Declination and its first time derivative time-series. Example for a high-standards geomagnetic observatory (Niemegk, NGK).**



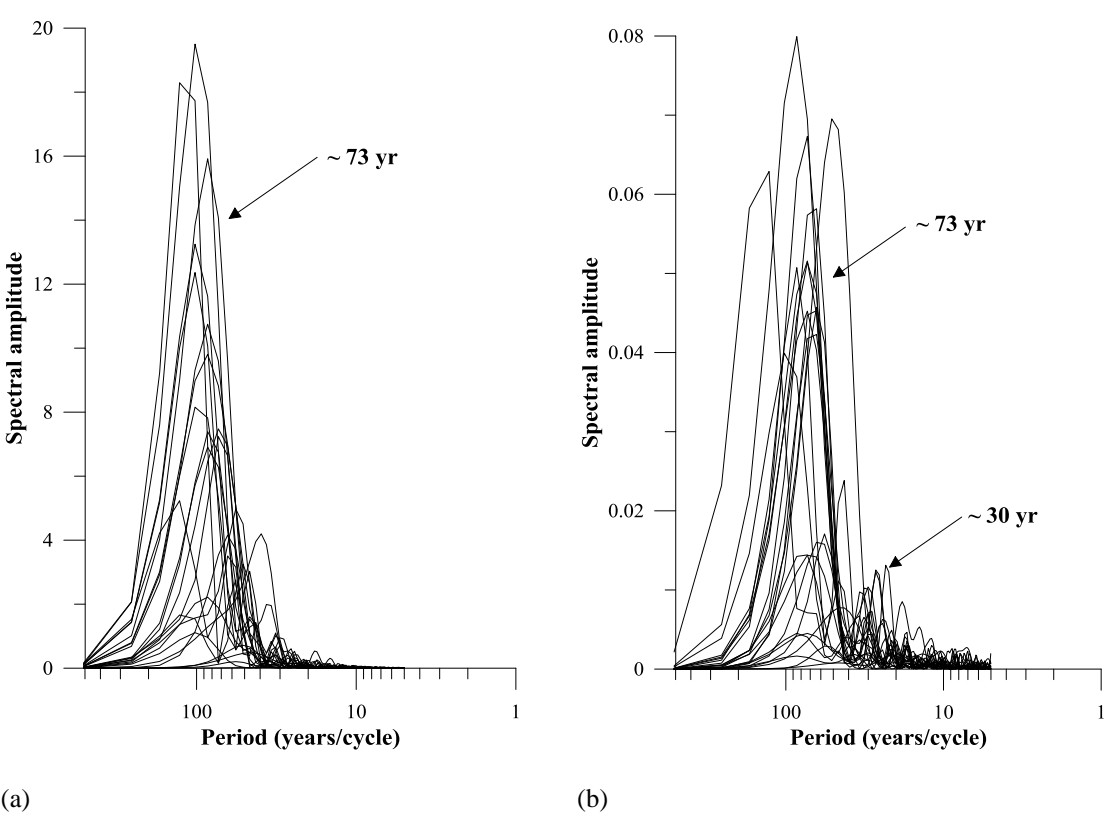

(a)                                                           (b)

**Figure 2: FFT power spectrum: observatory declination time-series (a); time derivative of declination time-series (b).**





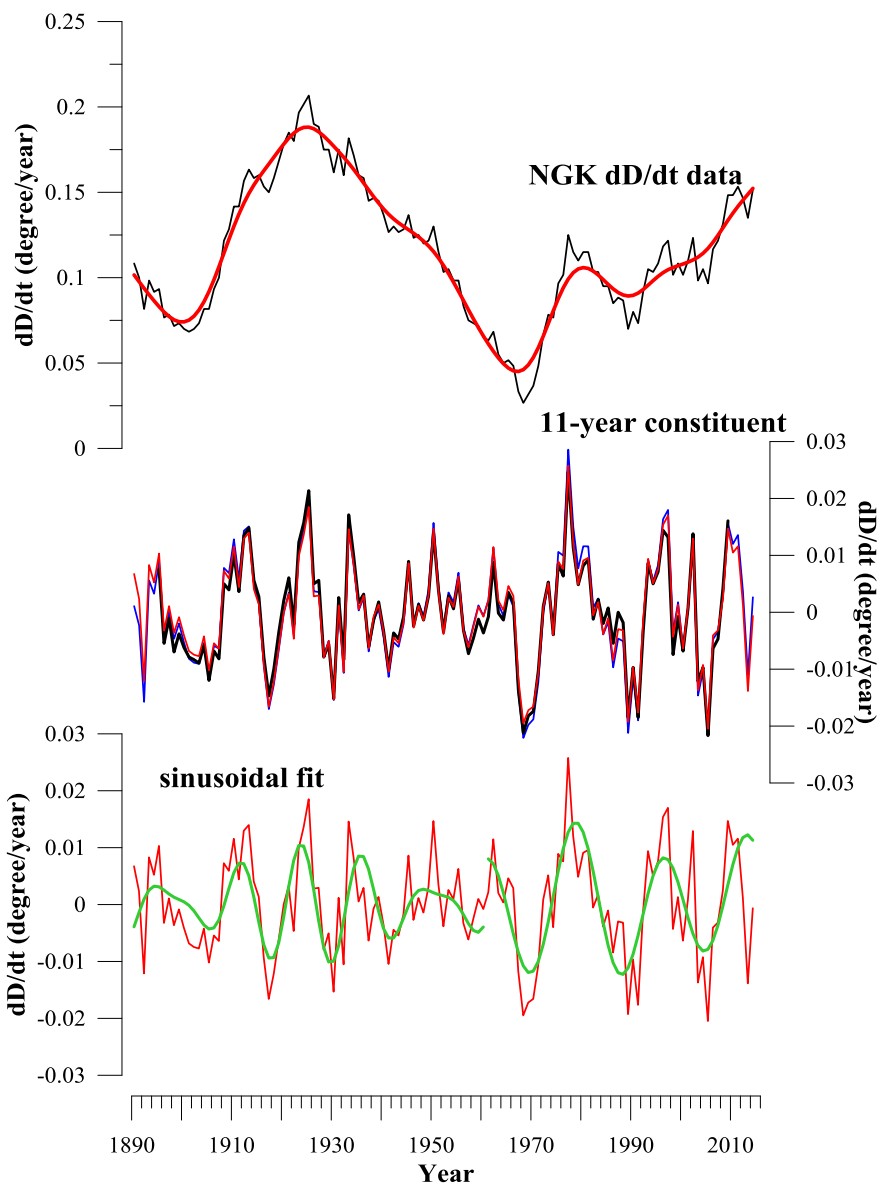

**Figure 3: Constituents of the first time derivative of the declination at Niemegk observatory. Top: first differences of annual means (thin black curve) and the trend from a HP filtering (thick red curve). Middle: cyclic constituent from a HP filtering (red), the 11-year signal from filtered first differences by an 11-year cutoff high-pass Butterworth filtering (blue), and the 11-year signal obtained as a residual after the removal of an 11-year running average from the dD/dt time-series (black). Bottom: the cyclic component (red) compared to a two-waves sinusoidal fit (green) for the time intervals 1890-1960 and 1960-2011.**




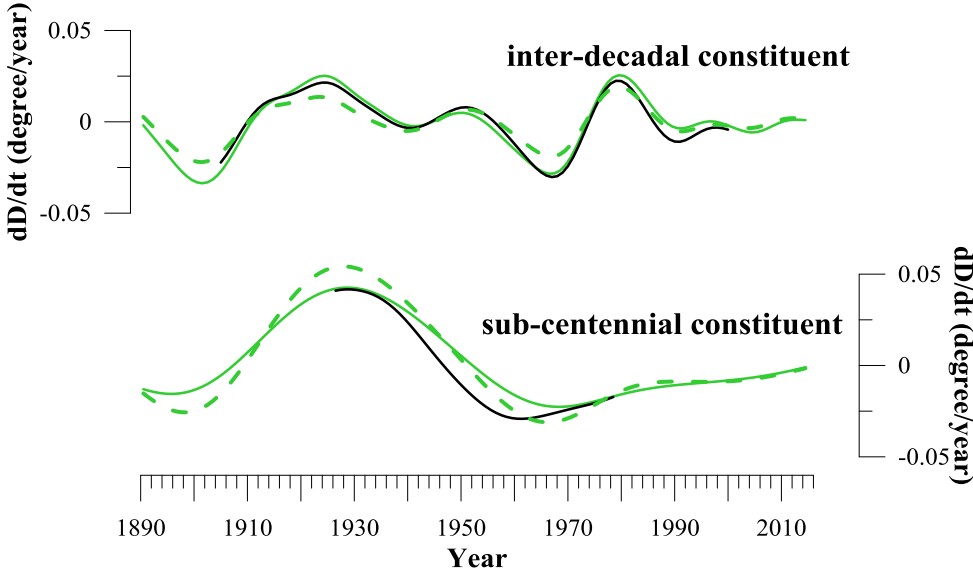

**Figure 4: Constituents of the first time derivative trend of declination at Niemegk observatory (the red curve in upper plot, Fig. 3).**

10  **Top: the inter-decadal constituent of the trend as a 30-year cutoff high-pass Butterworth filtering (green) and as a residual after removal of a 30-year running average filtering (black) from the trend, the '~22-year' constituent from a 22-year cutoff high-pass Butterworth filtering (dashed green). Bottom: the sub-centennial constituent of the trend as a 73-year cutoff Butterworth filtering (green), and as a residual after removal of a 73-year running average filtering from the 30-year smoothed trend (black), the '~80-year' constituent from a 78-year cutoff Butterworth filtering (dashed green).**



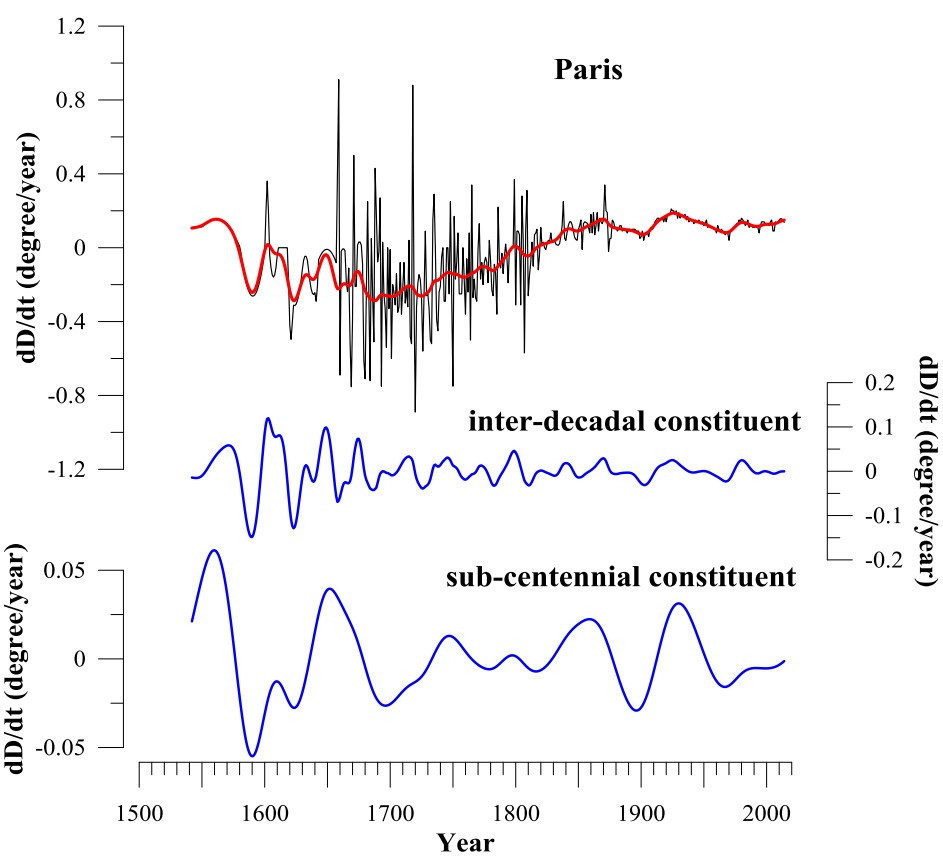

**Figure 5: The first time derivative of the Paris declination time-series (black) and the HP trend (red) (top panel); the inter-decadal (middle panel) and the sub-centennial (bottom panel) constituents of the trend from a Butterwoth filtering.**



**Figure 6: Constituents of the first time derivative of the declination at all analyzed observatories (European (black) and non-European (gray) time-series). From to bottom: the trend from a HP filtering, the cyclical constituent from a HP filtering, the inter-decadal constituent of the trend, the sub-centennial constituent of the trend. Vertical lines mark the generally accepted 20th century geomagnetic jerks.**



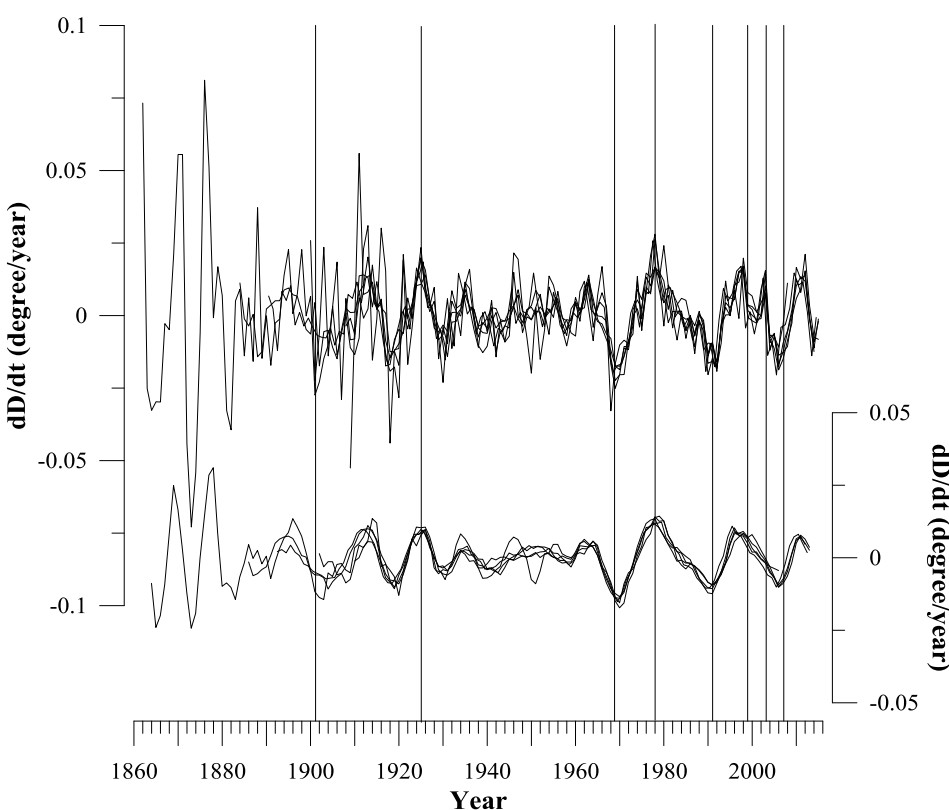

5  **Figure 7: Upper panel: the decadal variation of the first time derivative of declination at the European observatories (cyclical constituent from a HP filtering). Lower panel: the 5-year running averages smoothing of time-series plotted in the upper panel. Vertical lines mark the generally accepted 20[th] century geomagnetic jerks.**



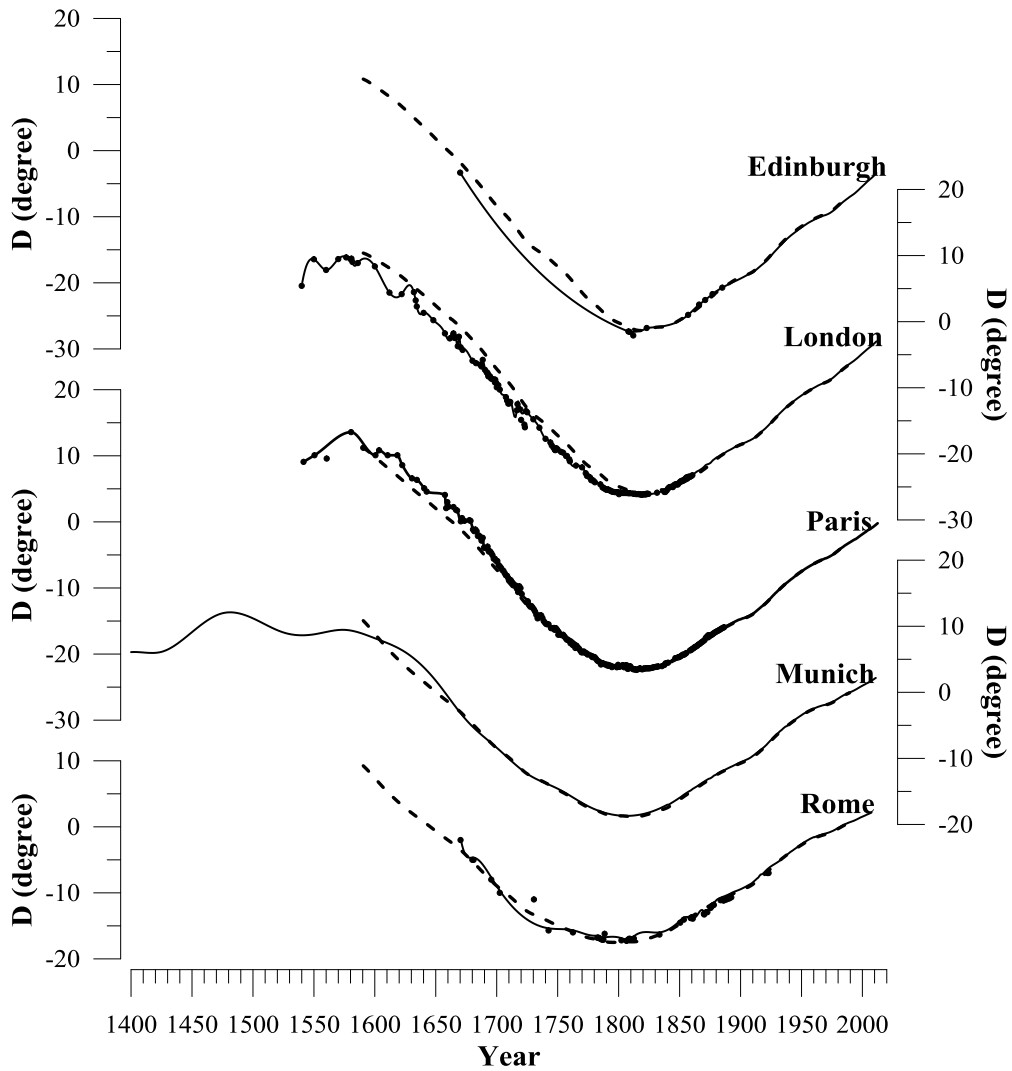

**Figure 8: The declination evolution since 1400 AD at five locations in Europe. Annual values as obtained by a cubic B-spline**

5 **interpolation on historical observations and observatory data (full black curve); and as obtained from the *gufm1* time-series (dashed curve).**





**Figure 9: The first time derivative of the HP trend in data (red) and in the *gufm1* (dashed black) for the five locations in Europe.**



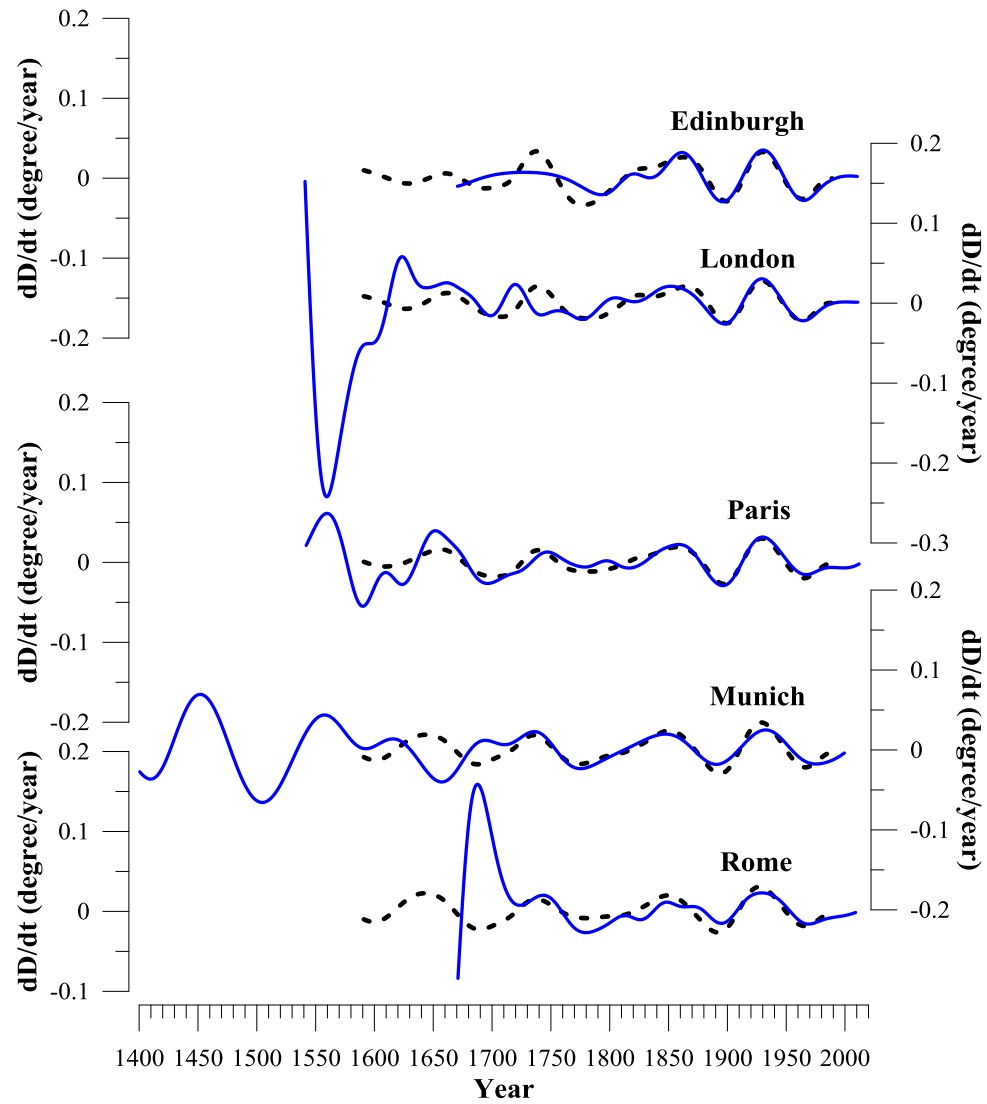

**Figure 10: The first time derivative of the sub-centennial constituent at the five locations, from the trend of observed data (blue) and of *gufm1* (dashed black); a high-pass 73-year cutoff Butterworth filtering on time-series of Fig. 9 is applied.**

