# Peer review of "Geomagnetic field declination: from decadal to centennial scales"

_Solid Earth, 2017_

## Referee Comment (RC1) · S. Macmillan (Referee) · 23 Nov 2017

This paper analyses various long series of declination data with the view to finding periodic signals and better revealing geomagnetic jerks. The techniques applied are Fast Fourier Transforms and visual viewing of traces after application of different filters whose pass bands are, in part, established from the FFT (Hodrick-Prescott, box-car, Butterworth).

There are some novel ideas about separating signals in geomagnetic data here and they have been applied to a new collection of geomagnetic data. However separating sources of the signals is not explored at all. The Hodrick-Prescott (H-D) filter is widely applied in business data where trends are being isolated in data with seasonal and

annual signals.

The conclusion "The generally accepted geomagnetic jerks occur around extrema in the time derivative of the trend" is hardly surprising as this is practically the definition of a geomagnetic jerk, insofar as we have one.

It is not clear to me that the conclusion "The generally accepted geomagnetic jerks...coincide with extrema in the time derivative of the 11-year constituent" can be obtained from Figures 6 and 7. These Figures show a coincidence of the jerk dates with extrema in various cyclic constituents, not just the 11-year cycle, of the time derivative. Your comment at the end about a definition for a jerk essentially says this though, viz "...the geomagnetic jerk concept should be considered as a more general notion, namely the evolution of the secular variation as a result of superposition of two (or more) constituents describing effects of processes in the Earth's core at two (or more) time-scales."

More detail is needed on how the 30-year and 73-year spectral lines are identified in Figure 2 because these values are then used (at least initially) in some of the subsequent filter design. Does the length of the series, which you comment is important, somehow weight the times of the observatory peaks in the averaging process?

In your description of the H-D filter are the expected periodicities part of the filter design? What are the advantages of the H-D filter over fitting a cubic spline then doing a spectral analysis to the residuals?

What is the reason for the split in the 3rd panel of Figure 3? NGK runs continuously through this time.

Please clarify how you get a $\sim$22-year oscillation over the past $\sim$40 years from Figure 3. It looks more like 16-17 years and in any case must be related to the 2 periods used in the "two-waves" sinusoidal fit.

An analysis of the sunspot series (a completely independent series) would be useful to

illustrate the significant departures from the 11-year cycle. This would also lend weight to the assertion that the high frequency cyclic signal is external in origin.

How can you be sure that the big jerks are not influencing the results of the filtering in Figure 6? This is crucial as your main conclusion seems to be along the lines of "geomagnetic jerks are not random features but a cyclical feature of Earth's magnetic field". An important conclusion potentially.

---

## Referee Comment (RC2) · J. Mound (Referee) · 7 Dec 2017

This work provides some new insight into the nature of geomagnetic variations over long (decadal to centennial) time scales, particularly with regards to apparent periodicities within observational time series of declination variations. This expands upon previous work by this team (and others) on whether such periods can be cleanly identified from observatory records, a task made difficult by the relatively short length of the continuous observatory time series (at best ~150 years) compared to the suspected periods (~30 and ~80 years). There are also difficulties in differentiating between periodic signals of internal origin from external effects, predominantly linked the well-established solar variations with typical periods of 11 years. To attempt to disentangle these components of observed magnetic variation, the authors apply a variety of different filters to 24 long geomagnetic observatory records from around the world (albeit with an unavoidable geographic bias), in particular making use of Hodrick-Prescott filtering, which to my knowledge has not been used in this context before. They also consider five very long (multi-century) reconstructions of declination variation from locations within Europe.

The results from the observatory records were quite similar to previous analyses, giving confidence in the application of HP filtering to these records, although, as in previous studies, it is very difficult to find cleanly separated and identified periods. I found the indication that the sub-centennial constituent is a coherent variation over the last several centuries particularly intriguing. However, since the five locations considered are all European, it may not be surprising that they show a similar variability, particularly within gufm1 as the spatial resolution of that model does decrease significantly towards the start of that model.

A minor addition regarding the discussion on page 3 lines 9-11 on the six-year geomagnetic signal, it should be noted that there is a corresponding six year signal in length-of-day that cannot be explained by known external sources and thus has been linked to processes in the core (this is discussed in the cited work by Gillet et al. and Home & de Viron; but see also work by Abarca del Rio et al., 2000, Annales Geophysicae; Mound and Buffett, 2006, EPSL; Duan et al., 2018, EPSL). Thus, although the mechanism responsible for the 6-year geomagnetic signal remains under debate it is almost certainly internal in origin.

Additionally, from a structural point of view, I might move that discussion so that it connects to the paragraph on page 2 lines 20-30; which ends with a discussion of these sort of inter-annual signals. The discussion on page 3 could then remain focused on the (inter)decadal and longer variations.

When applying the Butterworth filter, is there a reason to do this to the HP-filtered trend rather than applying it to the data directly? Perhaps running a filter on a filter doesn't

make any difference in this case as all of the removed signal is at frequencies much higher than those of interest, but even if that is the case, why not simply apply the Butterworth filter to the original time series?

On page 10 lines 20-23, there is a discussion of the relative timing of geomagnetic events (which may or may not be jerks) relative to the maxima and minima of the sub-centennial variation. The two dates are said to correspond "within a few years", but in reality there is about a decade between them. Given the uncertainty on both sides of this comparison, I might be more conservative in discussing the closeness of this correspondence.

Page 12, lines 3-5. I agree that after your processing to remove suspected external signals, the time series no longer has the sharp V-shaped events, that are often used as an identifier of jerks. However, your method of removing suspected external signals is essentially to filter out high frequencies, which necessarily results in smoother time series. If you applied your filters to a "perfect" V shaped or sawtooth function with a period of 20 years, what would survive? I suspect that it would also end up looking rather smooth. I don't see any easy resolution to this problem through pure time series analysis, comparison to external field models or proxies (e.g. indices of solar activity) seem necessary to unravel the origin of the high frequency content within the geomagnetic observatory time series.

Why is the green line in figure 3 discontinuous? Presumably this reflects a suspected change in frequencies that contribute to this signal. If this is truly external signal is there any corresponding change known of in measures of the external field?

In figures 9 & 10 the amount of noise in the data and the mismatch between the data and gufm1 appears to grow significantly between about 1700 and 1800 (the exact timing of this differs between sites). Therefore I might be cautious about claiming that the sub-centennial signal is really traced all the way back to 1600.

Other minor points:

I find the lightly weighted lines (particularly the grey lines in figure 6) very hard to see in printed form, although they are ok when viewed on the computer.

last line of page 2: "seemed to characteristic in case of declination" reads oddly to me, perhaps simply "characteristic of declination" would work better.

---

## Author Comment (AC1) · 20 Dec 2017

The comment was uploaded in the form of a supplement:
https://www.solid-earth-discuss.net/se-2017-119/se-2017-119-AC1-supplement.zip

---

## Author Comment (AC2) · 20 Dec 2017

The comment was uploaded in the form of a supplement:
https://www.solid-earth-discuss.net/se-2017-119/se-2017-119-AC2-supplement.zip

---

## Editor Comment (EC1) · N. Gillet (Editor) · 21 Dec 2017

Dear authors,

have you posted the revised manuscript ? I can only see the response to the referees...

Best Regards,
* * *

---

## Editor Comment (EC2) · N. Gillet (Editor) · 21 Dec 2017

yes, I need it to check that you indeed have modified the manuscript according to your reply.

regards.

---

## Author Comment (AC3) · 21 Dec 2017

Dear Editor, We posted only the response to the referees, not the revised manuscript. Should be post it, now?

Sincerely yours, Venera Dobrica
* * *

---

## Editor Comment (EC3) · N. Gillet (Editor) · 22 Dec 2017

would it be possible for you to post a pdf (instead of a doc file) since some figures do not show up properly ?

best regards
* * *

---

## Author Comment (AC4) · 22 Dec 2017

Dear Editor, Please find the revised manuscript uploaded as supplement. Sincerely yours, Venera Dobrica

Please also note the supplement to this comment:
https://www.solid-earth-discuss.net/se-2017-119/se-2017-119-AC4-supplement.zip

---

## Editor Comment (EC4) · N. Gillet (Editor) · 8 Jan 2018

Dear authors,

I have been through your response and the revised manuscript. I find that if you answer to several of the concerns raised by the referees, you sometimes pass too quickly over some of their comments (see point 3 below). Furthermore, it seems to me that some fundamental literature is ignored (point 1), and that synthetic tests should be carried out for the reader to accept your results (point 2). Consequently, I ask you to bring important modifications to your manuscript before it can be published to Solid Earth. I understand that it requires some significant work, but it is needed for the reader to accept your findings.

Best Regards

1) Part of the literature that cannot be ignored. For instance in the abstract you assume that decadal changes are « supposed to be related to external variations and called '11-year constituent' ». This is part of your background hypothesis, but you must recognize that this point of view cannot entirely be correct. Of course there exist an external signal at such periods; however, internal signals are of primary importance at these very periods, in link with the red spectrum of the internal field (Lesur et al, PEPI 2017).

Furthermore, the ambiguity in the internal/external sources separation being much reduced with satellite data, the community is pretty confident in a large signal from the core at periods around 11 yrs (see e.g. the global models such as CHAOS-6, Finlay et al, EPS 2016).

2) You should explain how your results (the existence of "cyclic constituents") is compatible or at odds with the approximately -4 slope found in the PSD of ground observatory series (de Santis et al, PEPI 2003; Lesur et al 2017). Indeed, series characterized by such a PSD slope in the considered frequency range do not contain any specific spectral line. Can you prove that the cyclic behavior you see (in particular towards long periods) do not come from the restricted time-span of the series ?

Said differently, you should apply your signal analysis tool to short segments of stochastic series presenting a PSD with a -4 slope over the considered frequency range ; if there you find cyclic constituents, it means they are possibly due to the limited duration of geophysical series. My concern is motivated by the fact that apparent periodicities are often wrongly put forward when too short series are considered.

3) Below is a list of comments on your responses to the referees (in blue the referees points, in red your response or modified text, in black my comments) :

on comments by Susan Macmillan

3.1- More detail is needed on how the 30-year and 73-year spectral lines are identified in Figure 2 because these values are then used (at least initially) in some of the subsequent filter design...

Most observatories indicate these figures for the spectral line corresponding period… the filtered time series do not lose the information related to actual periods involved, no matter what figure is used in filtering.

You do not answer the question : how do you estimate the periods ? Is it by applying an average over the several spectra ? Could you provide errorbars on the obtained period? Behind the question of the referee, I understand that the periods are not obvious from fig. 2.

3.2- What is the reason for the split in the 3rd panel of Figure 3? NGK runs continuously through this time. (a point also raised by J. Mound)

… We treated separately the two parts of the signal (1890-1960 and 1961-2014) because the plot suggests a change in frequency that contribute to the signal…

This affirmation is not obvious at all to me, as in both parts of the series it only concerns a few periods. Furthermore, the accuracy you give for the periods seems illusory to me. I do not think any conclusion such as "beatings between the sunspot (so-called 11-year) and magnetic (~22-year) solar cycles" can be drawn. Here analyses of synthetic series (see point 2) would be useful. Indeed, stochastic series often show natural modulations that look like changes in apparent periods (although no period line actually exists).

3.3- How can you be sure that the big jerks are not influencing the results of the filtering in Figure 6? … an important conclusion potentially.

The results of filtering are not influenced by the position of the big jerks.

This is not an answer. Could you detail a bit ? Again you could illustrate this with synthetic tests.

On comments by Jon Mound:

3.4- in your modifications following the comment "… could be noted that there is a corresponding six year signal în length-of-day that cannot be explained by known external sources and thus has been linked to processes in the core…" you refer to Cox et al (2016) saying "… shows, however, that a 6-year wave in the core cannot give the estimated effects at Earth's surface, placing the problem of internal high-frequency signals under debate."

This is not correct. Indeed, what Cox et al show is that the 6 yr signal from synthetic geostrophic waves (with amplitude that found by Gillet et al 2010) is tiny, comparable with the uncertainty level in observatory series. However, around 6 yr periods, core flows inverted from magnetic data are dominated by more intense non-geostrophic motions that are able to explain the resolved signal at such periods (i.e. there is no problem for interannual magnetic signals to be explained by core motions). Furthermore, because there are about 100 independent observatories to constrain the secular variation, the uncertainty level on core motions is much reduced. This explains why such small geostrophic motions can be retrieved even if they are only responsible for a tiny signal.

3.5- You do not modify the text in response to "… I don't see any easy resolution to this problem through pure time series analysis, comparison to external field models or proxies (e.g. indices of solar activity) seem necessary to unravel the origin of the high frequency content within the geomagnetic observatory time series.". You write "Once the external contributions to the first differences of the observatory annual means – of comparable amplitude with the inter-decadal and sub-centennial constituents – are minimized, the observed secular variation no longer exhibits a clear V-shape at time of geomagnetic jerks." Following point 1 above, how can you be sure you have only removed external signals ? In your conclusion you should acknowledge that you have most probably also removed some important internal signal. This may be part of the reason why the correlation with of the 11 yr constituent with the solar cycle is not so clear. You should also mention somewhere the attempts at extracting external contributions through global models on long periods (McLeod et al, JGR 1996; Langel et al, PEPI 1996; Gillet et al, G3 2013), who give an idea of what wan be achieved on the basis of spherical harmonics decomposition, and of the expected respective amplitudes of internal and external signals.

3.6- … Therefore I might be cautious about claiming that the sub-centennial signal is really traced all the way back to 1600.

… As the sub-centennial variation in *gufm1* closely follow the observed time series in the last ~200 years of the time series depicted in Fig. 10, there is ground to give credit to the entire time series.

I don't see the logic of your response. The point of the referee remains valid, and should be explicitly acknowledged.

---

## Author Comment (AC6) · 18 Jan 2018

Dear Editor, Could you kindly tell me the deadline for sending the revised manuscript according to your comments? Thanking you in advance, Yours sincerely, Venera Dobrica

---

## Editor Comment (EC5) · N. Gillet (Editor) · 22 Jan 2018

Let say one month, the 23rd of debruary. best regards.
* * *

---

## Author Response (AR1)

Dear Editor,

Thank you for the letter regarding the paper SE-2017-119. We revised our manuscript taking into account your observations, as follows.

Sincerely,

5  Venera Dobrica, Crisan Demetrescu, Mioara Mandea

1) Part of the literature that cannot be ignored. For instance in the abstract you assume that decadal changes are « supposed to be related to external variations and called '11-year constituent' ». This is part of your background hypothesis, but you must recognize that this point of view cannot entirely be correct. Of course there exist an external signal at such periods;
10  however, internal signals are of primary importance at these very periods, in link with the red spectrum of the internal field (Lesur et al, PEPI 2017).
Furthermore, the ambiguity in the internal/external sources separation being much reduced with satellite data, the community is pretty confident in a large signal from the core at periods around 11 yrs (see e.g. the global models such as CHAOS-6, Finlay et al, EPS 2016).

We agree that our point of view regarding the signal at frequencies related to the solar activity as an external one might not be correct. We tend to attribute the 11-year signal to the external sources (our background hypothesis) based on a long list of papers referenced by Demetrescu and Dobrica (PEPI 2014) and on the demonstrations by Verbanac et al. (EPS 2007) and Wardinski and Holme (GJI 2011),
20  to cite a few. The former showed that the residual signal after removing the CM4 core field contribution from the annual averages of the European observatories has a clear solar-cycle signature and can be modeled down to ±2 nT using magnetospheric ring current data (Dst) and an ionospheric field proxy (Ap). The latter characterizes the external effects stochastically, analyzing the correlated so-called 'noise' in the time series of X, Y, and Z at observatories world-wide and the Dst index, in a first step,
25  and residuals at Niemegk instead of Ap, in a second step. Moreover, there is no published information on the amplitude (or spectral power density) of the internal 11-year signal that we could mention here, as a possible alternative of our working hypothesis.

We agree that models based on satellite data succeeded in drastically reducing the ambiguity in the external/internal sources separation. However, the information from double logarithmic plots of FFT
30  spectra of, e.g. Lesur et al. (2017), only shows that there is an 11-year signal in the **core field of the model**, but cannot decide on the origin. The same has been noted by De Santis et al. (PEPI 2003): "The observational spectrum follows the predicted linear behaviour relatively well down to periods of about 7 years; we do not know whether this is simply fortuitous or whether it depends on possible external contamination to annual means at the shortest periods or it has some other physical meaning". A
35  paragraph mentioning the above studies have been included in Introduction (page 2, lines 33-34, page 3, lines 1-3) of the revised manuscript.

**Perhaps information on the amplitude (or spectral power density) of the internal 11-year signal, which is not discussed in the currently published papers, would give a definitive answer.** For the moment we added in the revised manuscript (Page 6, lines 20-22; page 13, lines 11-12) that we
40  are aware that our point of view regarding the signal at frequencies related to solar activity as an external one might not entirely be correct.

2) You should explain how your results (the existence of "cyclic constituents") is compatible or at odds with the approximately -4 slope found in the PSD of ground observatory series (de Santis et al, PEPI 2003; Lesur et al 2017). Indeed, series characterized by such a PSD slope in the considered frequency range do not contain any specific spectral line. Can you prove that the cyclic behavior you see (in particular towards long periods) do not come from the restricted time-span of the series ?

Said differently, you should apply your signal analysis tool to short segments of stochastic series presenting a PSD with a -4 slope over the considered frequency range ; if there you find cyclic constituents, it means they are possibly due to the limited duration of geophysical series. My concern is motivated by the fact that apparent periodicities are often wrongly put forward when too short series are considered.

1. Our filtering is based on well documented findings of a long line of preceding researchers, listed in the references, that previously studied periodical signals in geomagnetic data (we added several lines (23-26) at page 4 concerning the paper by Jackson and Mound (2010) who showed by EMD periods that are in line with ours). We do not use filtering to find periodicities, but rely on already found ones.

2. The figure below shows a 400 points (=years) stochastic time series with the approximartely -4 slope of its PSD plus the corresponding spectrum in log-log form. Also spectra for shorter time series (200 –blue, 100 years –red) cut from the initial one are superimposed. We do not see spurious lines at low frequencies in case of shorter time series.

[Figure]

Stochastic time series (black, upper panel), trend get by HP filtering (red, upper panel) and cyclic constituent (black, lower panel)

[Figure]

FFT power spectra in log-log representation for 400- (black), 200- (blue) and 100-years (red) stochastic time series

3. The superimposed power spectra of our 24 declination time series of secular variation (dD/dt) are shown in the figure below, together with the mean spectrum (red) and two straight line segments (green) illustrating the power law that characterize the mean spectrum; a slope change occurs at a period of about 10 years. The power-law exponent is -1.81, similar to the figure found by Lesur et al. (2017) for individual observatories.

[Figure]

The log-log power spectra (black) and the mean spectrum (red) of declination secular variation with the power-law fit (green)

4. The same analysis on declination time series (D) results in almost the same exponent (-1.89) – see figure below, but not close to -4 as got by De Santis et al. (2003).

[Figure]

The log-log power spectra (black) and the mean spectrum (red) of declination with the power-law fit (green)

4.    Our dD/dt log-log plot shows „lines", or rather, periods grouped in three time-scales, namely 60-90, 20-35, and 8-15 years, exactly those we considered for our filtering approach. The actual figure we use for filtering window or cutoff does not alter the frequency content of the filtered time series (Demetrescu and Dobrica, PEPI 2014, Appendix). A spectral density power versus frequency or period plot is able to comparatively show the magnitude of various oscillations present in the time series, and it is only to be expected that a log-log plot smoothes the spectral lines, resulting in the power law one finds.

We added a new figure in the Supplementary material (FIG. S2) to show how our results are compatible or at odds with the approximately -4 exponent of the power-law fitted to spectra and a paragraph in the revised manuscript (page 4 at lines 26-30 and page 5, lines 1-5).

3) Below is a list of comments on your responses to the referees (in blue the referees points, in red your response or modified text, in black my comments) on comments by Susan Macmillan

3.1- More detail is needed on how the 30-year and 73-year spectral lines are identified in Figure 2 because these values are then used (at least initially) in some of the subsequent filter design...
Most observatories indicate these figures for the spectral line corresponding period… the filtered time series do not lose the information related to actual periods involved, no matter what figure is used in filtering.
You do not answer the question : how do you estimate the periods ? Is it by applying an average over the several spectra ? Could you provide errorbars on the obtained period? Behind the question of the referee, I understand that the periods are not obvious from fig. 2.

Indeed, we applied an average over the 24 spectra, as De Silva et al. (JGRB 2012, Fig. 2), Demetrescu and Dobrica (PEPI 2014, Fig.3), and Ou et al. (JGRA 2017, Fig. 3) proceeded with their data. What is important, however, is that no matter what figure was used in our filter design (except the actual period in data), the filtered time series would show the actual oscillations hidden in the unfiltered time series (Demetrescu and Dobrica, PEPI 2014, Appendix). We added that in the 2nd round revised manuscript (Page 5, lines 28-31).

3.2- What is the reason for the split in the 3rd panel of Figure 3? NGK runs continuously through this time. (a point also raised by J. Mound)
… We treated separately the two parts of the signal (1890-1960 and 1961-2014) because the plot suggests a change in frequency that contribute to the signal…
This affirmation is not obvious at all to me, as in both parts of the series it only concerns a few periods.
Furthermore, the accuracy you give for the periods seems illusory to me. I do not think any conclusion such as "beatings between the sunspot (so-called 11-year) and magnetic (~22-year) solar cycles" can be drawn.

Indeed, there are only a few periods in each of the two parts of the series. We agree with your comment regarding the accuracy we gave for the periods and about our conclusion. We removed that part of the manuscript (including the plot in question). As a matter of fact, external variation in D is rather different when compared to H and Z, due perhaps to additional sources that contribute, beside the symmetric ring current. We leave this matter for a next study.

Here analyses of synthetic series (see point 2) would be useful. Indeed, stochastic series often show natural modulations that look like changes in apparent periods (although no period line actually exists).
We do not consider our time series as stochastic but rather a result of turbulent flow in the outer core. We had already stated in the manuscript (page 9, line 32-35, pag. 10, lines 1-2 in the revised one): "That the flow in the core is turbulent became common consideration with many studies on geodynamo and core flow modelling from secular variation data (see the review by Holme (2015) on the latter). The turbulent flow, as inferred by De Santis et al. (2003) by looking at the power spectra of X, Y, Z annual means time-series, does not exclude but, on the contrary implies the existence of vortices with various time and space scales."

3.3- How can you be sure that the big jerks are not influencing the results of the filtering in Figure 6?… an important conclusion potentially.
The results of filtering are not influenced by the position of the big jerks.
This is not an answer. Could you detail a bit ? Again you could illustrate this with synthetic tests.

We give here a more detailed and convincing answer. We were missled by the term 'big jerk', as successive jerks have about the same magnitude.

We have previously noticed that dominating powerful signals at larger time-scales in data tend to contaminate the filtered time-series meant to show quasi-periodic variations at smaller time-scales, when using running window and Butterworth filtering (as stated in the present revised manuscript at page 5, lines 9-10). This was the reason for which we adopted the HP filtering, to avoid this happening in case of the 11-year variation. In Fig. 6 there is an obvious effect of the variation at the 60-90 year scale on the inter-decadal variation. It is the result of using Butterworth filtering on trend time series to get the inter-decadal and the sub-centennial constituents. The position in time of maxima and minima of the inter-decadal constituent can, however, stand for jerk moments, as the filtering does not influence the jerks timing.

On comments by Jon Mound:

3.4- in your modifications following the comment "… could be noted that there is a corresponding six year signal în length-of-day that cannot be explained by known external sources and thus has been linked to processes in the core…" you refer to Cox et al (2016) saying "… shows, however, that a 6-year wave in the core cannot give the estimated effects at Earth's surface, placing the problem of internal high-frequency signals under debate."

This is not correct. Indeed, what Cox et al show is that the 6 yr signal from synthetic geostrophic waves (with amplitude that found by Gillet et al 2010) is tiny, comparable with the uncertainty level in observatory series. However, around 6 yr periods, core flows inverted from magnetic data are dominated by more intense non-geostrophic motions that are able to explain the resolved signal at such periods (i.e. there is no problem for interannual magnetic signals to be explained by core motions). Furthermore, because there are about 100 independent observatories to constrain the secular variation, the uncertainty level on core motions is much reduced. This explains why such small geostrophic motions can be retrieved even if they are only responsible for a tiny signal.

We considered your comment. In our reasoning we were led by our background hypothesis that the 11-year signal, and consequently other, higher frequency signals are of external origin, as has been demonstrated by Ou et al. (2017) for the case of QBO-related variations. In the present revised manuscript we removed our comment on the Cox et al. paper.

3.5- You do not modify the text in response to "… I don't see any easy resolution to this problem through pure time series analysis, comparison to external field models or proxies (e.g. indices of solar activity) seem necessary to unravel the origin of the high frequency content within the geomagnetic observatory time series.". You write "Once the external contributions to the first differences of the observatory annual means – of comparable amplitude with the inter-decadal and sub-centennial constituents – are minimized, the observed secular variation no longer exhibits a clear V-shape at time of geomagnetic jerks."

Following point 1 above, how can you be sure you have only removed external signals ? In your conclusion you should acknowledge that you have most probably also removed some important internal signal. This may be part of the reason why the correlation of the 11 yr constituent with the solar cycle is not so clear. You should also mention somewhere the attempts at extracting external contributions through global models on long periods (McLeod et al, JGR 1996; Langel et al, PEPI 1996; Gillet et al, G3 2013), who give an idea of what wan be achieved on the basis of spherical harmonics decomposition, and of the expected respective amplitudes of internal and external signals.

Of course separation between internal and external sources is far to be perfect. We made the suggested changes in the 2nd round revised manuscript: page 6, lines 12-23 and pag.13, lines 11-12.

3.6- … Therefore I might be cautious about claiming that the sub-centennial signal is really traced all the way back to 1600. … As the sub-centennial variation in *gufm1* closely follow the observed time series in the last ~200 years of the time series depicted in Fig. 10, there is ground to give credit to the entire time series.
I don't see the logic of your response. The point of the referee remains valid, and should be explicitly acknowledged.

We added comments in the revised text, arguing for the presence of the sub-centennial variation back to 1600 (page 11, lines 2-7 and lines 18-19).

We state that the sub-centennial variation can be traced back to 1600, meaning that the corresponding wave was singled out in the reconstructed and *gufm1* time series by the filtering procedure used, as shown in Fig. 10 (Fig. 9 shows, in red, time series of observed data, that are indeed noisy in the first 100 years). The sub-centennial signal is noisier at the beginning of the five time series and mismatches to *gufm1* are evident at 1650 and 1700 for Munich and Rome. However, the *gufm1* time series are consistent with each other. On the oher hand, *gufm1* is not based only on the five time series, but on a much larger number of measurements and the SH analysis distributes errors over the entire spherical surface, so perhaps *gufm1* is able to satisfactory represent the sub-centennial variation at the beginning of the time series. We remind that the sub-centennial variation is the outcome of the filtering **only if** the time series subjected to filtering contains that variation.

[revised manuscript text omitted]

(a)

[Figure]

5    (b)

**Figure S2: FFT power spectrum in a log-log representation: (a) observatory declination time series; (b) declination time derivative time series. The average power spectrum (red) and the best power-law fit (green).**

[Figure]

**Figure S3: FFT power spectrum for the HP trend (i.e. the ~11-year variation removed) of declination first time derivative. The average power spectrum (red).**

[Figure]

**Figure S4: FFT power spectrum for the cyclic constituent from HP filtering of the first time derivative of declination. Gray background: domains of 8-14 years, 15-25 years, 4-7 years and 2-3 years.**

---

## Editor Decision (ED1)

Comments on the authors' response to the comments on the manuscript SE-2017-119 (Dobrica et al).

If the authors have answered almost all concerns raised in the previous iterations, there still remain a few modifications needed, before their paper can be accepted for publication to Solid Earth :

1. It seems their is some ambiguity with the two first figures in the authors response. The authors mention a synthetic series with a -4 slope PSD… I guess the 1st figure in their response is for the time derivative of the series ? Which would be coherent with the approx. -2 slope PSD on the FFT (page 3 of their response). Am I correct ?

2. In the corrected version, it seems there is some confusion on the top of page 2. The authors say

   "the temporal spectra of the geomagnetic field at the Earth's surface and of the core field spherical harmonic coefficients could be approximated by a power law with a negative slope of about -4… and of about -2, respectively. The latter succeeded to reach periods down to 1 or 2 years."

   But the -2 slope in Lesur et al (2017) is for the SV ! Which corresponds to -4 for main field series. So the two studies (by de Santis and by Lesur) are coherent ! The latter indeed extending this result down to shorter periods. The text should be corrected accordingly.

3. On the figures S2 (pages 4 and 5 of the response) : I do not understand how you can have a series dD/dt that gives a PSD with a slope about $s1 = -1.8$, together with the corresponding series D(t) giving a PSD with a slope about $s0 = -1.9$…. since one should have $s0=s1-2$. Such a value would make your analysis consistent with that of de Santis et al (2003) (and see point 2 above).

   I have the feeling your PSD for D is polluted by edge effects – see the many oscillations on the PSD for D ! These would disappear if removing the end-to-end line in the series (+ using tappers might help), the recipe used by de Santis et al. Note that this could also affect the PSD of dD/dt (to a lesser extend), and will make this analysis coherent with the result of de Santis et al (2003).

   Furthermore, the three period ranges highlighted by the authors (60-90, 20-35 and 8-15 yrs) do not look obvious to me at all when looking at the log-log representation of the PSD for dD/dt. Before considering such periods, and given the suspected edge effects, I would need to see the PSD for dD/dt computed while removing the end-to-end line + using a tapper.

   These issues should really be considered before publication, with modifications to the corresponding text at the beginning of section 3.

---

## Author Response (AR2)

Dear Editor,

Thank you for the letter regarding the paper  SE-2017-119. We revised our manuscript taking into account your comments. The changes in the manuscript have been done in green in our previous red paragraphs on pages 2-3 and 4-5.

Below are the authors responses (in red) at the questions arised by you.

Sincerely,

Venera Dobrica, Crisan Demetrescu, Mioara Mandea

1.It seems their is some ambiguity with the two first figures in the authors response. The authors mention a synthetic series with a -4 slope PSD… I guess the 1st figure in their response is for the time derivative of the series ? Which would be coherent with the approx. -2 slope PSD on the FFT (page 3 of their response). Am I correct ?

We checked our files: actually, the two figures correspond to a stochastic time series with a built in -2 slope of the log-log plot. It serves, however, the purpose of showing the effect shorter, cut time series have. In the two following figures we show (1) a stochastic time series with a built in slope of about -4 of FFT log-log and (2) the FFT log-log plots for the entire series (400 points, black), the last 200 points time series (blue), and the last 100 points (red), as well as the linear fit.

[Figure]

Stochastic time series with a built in PSD slope of about -4

[Figure]

FFT power spectra in log-log representation for 400- (black), 200- (blue) and 100-years (red) stochastic time series. The power law fit with an exponent of -3.88 (black straight line)

2. In the corrected version, it seems there is some confusion on the top of page 2. The authors say "the temporal spectra of the geomagnetic field at the Earth's surface and of the core field spherical harmonic coefficients could be approximated by a power law with a negative slope of about -4… and of about -2, respectively. The latter succeeded to reach periods down to 1 or 2 years."
But the -2 slope in Lesur et al (2017) is for the SV ! Which corresponds to -4 for main field series. So the two studies (by de Santis and by Lesur) are coherent ! The latter indeed extending this result down to shorter periods. The text should be corrected accordingly.

We corrected the text. It reads now (page 2, lines 33, 34, page 3 lines 1-6):

By analysing the frequency content of the geomagnetic field variability, De Santis et al. (2003) and Lesur et al. (2017) have been able to reveal the behaviour of the geomagnetic field as either chaotic or stochastic. The former showed that the temporal spectra of the geomagnetic field at the Earth's surface could be approximated, for the 1871-2000 time span, by a power law with a negative slope of about -4 (in the frequency range corresponding to periods from 7 to 64 years). The latter showed, on shorter time series covering the time interval 1957-2014 and on their model coefficients, that in case of SV the corresponding power law slope is of about -2, in accordance with De Santis et al. (2003) results and with Gillet et al. (2013) hypothesis in deriving the main field COV-OBS model. Also, Lesur et al. (2017) succeeded to reach periods down to 1 or 2 years.

3. On the figures S2 (pages 4 and 5 of the response) : I do not understand how you can have a series dD/dt that gives a PSD with a slope about s1 = -1.8, together with the corresponding series D(t) giving a PSD with a slope about s0 = -1.9…. since one should have s0=s1-2. Such a value would make your analysis consistent with that of de Santis et al (2003) (and see point 2 above).

I have the feeling your PSD for D is polluted by edge effects – see the many oscillations on the PSD for D ! These would disappear if removing the end-to-end line in the series (+ using tappers might help), the recipe used by de Santis et al. Note that this could also affect the PSD of dD/dt (to a lesser extend), and will make this analysis coherent with the result of de Santis et al (2003).

Furthermore, the three period ranges highlighted by the authors (60-90, 20-35 and 8-15 yrs) do not look obvious to me at all when looking at the log-log representation of the PSD for dD/dt.
Before considering such periods, and given the suspected edge effects, I would need to see the PSD for dD/dt computed while removing the end-to-end line + using a tapper.

These issues should really be considered before publication, with modifications to the corresponding text at the beginning of section 3.

Actually in our FFT analysis we detrend time series by removing a linear trend from data, but indeed, the power spectra are contamined by edge effects. By removing the end-to-end line from data these effects are much lesser visible in the corresponding power spectra. We removed the end-to-end line in the time series of D and dD/dt, reconstructed Fig. S2, and added in the revised text that our results confirm the De Santis et al. (2003) and Lesur et al. (2017) findings (page 5, lines 1-3). We also added some details in the Method section (page 4, lines 22-30 and page 5, lines 1-7).

We reproduce below the revised dD/dt of Fig S2. In spite of the flatening that results from log-log plotting, one can still recognize the groups of lines.

[Figure]

The log-log power spectra (black) and the mean spectrum (red) of declination secular variation with the power-law fit (green)

[revised manuscript text omitted]